# The Geomagnetic Field (GMF) Is Required for Lima Bean Photosynthesis and Reactive Oxygen Species Production

**DOI:** 10.3390/ijms24032896

**Published:** 2023-02-02

**Authors:** Ambra S. Parmagnani, Nico Betterle, Giuseppe Mannino, Stefano D’Alessandro, Fabio F. Nocito, Kristina Ljumovic, Gianpiero Vigani, Matteo Ballottari, Massimo E. Maffei

**Affiliations:** 1Department of Life Sciences and Systems Biology, University of Turin, Via Quarello 15/a, 10135 Turin, Italy; 2Dipartimento di Biotecnologie, Università degli Studi di Verona, Strada le Grazie 15, 37134 Verona, Italy; 3Dipartimento di Scienze Agrarie e Ambientali—Produzione, Territorio, Agroenergia, Università degli Studi di Milano, 20133 Milano, Italy

**Keywords:** reaction center, chlorophyll, carotenoid, photosystems I and II, fluorescence kinetics, hydrogen peroxide, chloroplast ultrastructure, near null magnetic field, iron-sulfur complex assembly, magnetoreception

## Abstract

Plants evolved in the presence of the Earth’s magnetic field (or geomagnetic field, GMF). Variations in MF intensity and inclination are perceived by plants as an abiotic stress condition with responses at the genomic and metabolic level, with changes in growth and developmental processes. The reduction of GMF to near null magnetic field (NNMF) values by the use of a triaxial Helmholtz coils system was used to evaluate the requirement of the GMF for Lima bean (*Phaseolus lunatus* L.) photosynthesis and reactive oxygen species (ROS) production. The leaf area, stomatal density, chloroplast ultrastructure and some biochemical parameters including leaf carbohydrate, total carbon, protein content and δ^13^C were affected by NNMF conditions, as were the chlorophyll and carotenoid levels. RubisCO activity and content were also reduced in NNMF. The GMF was required for the reaction center’s efficiency and for the reduction of quinones. NNMF conditions downregulated the expression of the *MagR* homologs *PlIScA2* and *PlcpIScA*, implying a connection between magnetoreception and photosynthetic efficiency. Finally, we showed that the GMF induced a higher expression of genes involved in ROS production, with increased contents of both H_2_O_2_ and other peroxides. Our results show that, in Lima bean, the GMF is required for photosynthesis and that *PlIScA2* and *PlcpIScA* may play a role in the modulation of MF-dependent responses of photosynthesis and plant oxidative stress.

## 1. Introduction

The geomagnetic field (GMF) is an environmental component of Earth and, together with gravity, light, temperature and water availability, has always been present on our planet since the beginning of plant evolution [1]. The GMF protects life on Earth against the solar wind and cosmic radiation that would otherwise strip away the Earth’s atmosphere, including the ozone layer that protects the Earth from harmful ultraviolet radiation [2]. Moreover, the GMF could have influenced the origin [2] and evolution [3] of life. The average GMF amplitude is about 40 µT, but there are significant local differences in the strength and direction of the Earth’s magnetic field. At the surface of the Earth, the vertical component is maximal at the magnetic pole, amounting to about 67 μT, and is zero at the magnetic equator. The horizontal component is maximal at the magnetic equator, about 33 μT, and is zero at the magnetic poles [3]. In contrast with most other environmental factors, and similar to gravity, the GMF is present both at night and during the day, is largely unaffected by weather and seasons, and exists virtually everywhere on Earth. Thus, it is perhaps not surprising that diverse organisms, ranging from bacteria to vertebrate animals [4,5], have evolved ways to exploit the GMF to guide their movements [6].

The GMF acts on living organisms and influences many biological processes [7]. The leading hypothesis about a magnetic field (MF) receptor center on the radical pairs (RP) model suggests that the magnetic compass sense is based on reversible light-dependent chemical reactions [8,9]. Cryptochromes in plants undergo a light-dependent electron transfer reaction and have been shown to be involved in MF responses [10]. A linkage between cryptochromes and response to MF variations has been documented [7,10,11,12,13,14,15,16,17,18,19]. In photosystem II (PSII), the primary RP is formed by charge separation after excitation of the reaction center chlorophyll (Chl) P_680_ [20]. Iron-sulfur (Fe-S) complex assembly (IScA) scaffold proteins, which are an important component in the functioning of photosystems, are involved in abiotic stress responses. In animals, they act as magnetic sensors by forming a magnetoreceptor (MagR) complex with cryptochrome (Cry) [21]. The expression of *MagR* and *Cry* genes has been demonstrated in different organisms [22,23] and was evaluated in the model plant *Arabidopsis thaliana* [24]. In this plant, the GMF affects the expression of iron-uptake genes under iron deprivation conditions and the simultaneous alteration of Fe-S availability [25]. Moreover, alteration of the GMF intensity affects Fe content and homeostasis [26]. The Fe-S cluster can transfer electrons in electron transfer chains (e.g., in photosynthesis) and catalyze redox reactions [27]. Plastids, which are a major subcellular sink of iron and Fe-S clusters, are pivotal to photosynthesis in the electron transport (ET) chain [28]. Examples of chloroplastic Fe-S proteins include 2Fe-2S-containing chloroplast ferredoxin, 3Fe-4S-containing ferredoxin-glutamine oxoglutarate aminotransferase and 4Fe-4S-containing Photosystem I core proteins PsaA, PsaB, and PsaC ([29] and references therein). MF-stimulating effects have been described in regard to the evolution of a RP appearing in both photosystems I and II (PSI, PSII) [30,31] and were explained by a hypothesis of RP recombination in reaction centers [32]. Therefore, variations in the MF may affect the recombination of the primary RP to produce the triplet state of the primary donor (^3^P_680_), which can react with O_2_ to produce reactive singlet oxygen (^1^O_2_), which can damage PSII by inhibiting the repair of photoinhibited PSII [20]. The production of reactive oxygens species (ROS) is therefore one of the consequences of plant responses to GMF [33].

Based on the above considerations, we evaluated the photosynthetic responses to altered MF conditions in a crop plant (Lima bean) to assess the physiological, metabolomic and cytological responses. To study these effects, we built an exposure system made of a triaxial Helmholtz coils system of about 8 m^3^ able to reduce the GMF from values of about 42,000 nT and inclination of about 58° (average values in Italy) to Near Null MF (NNMF) values (about 30 nT) [34]. With this system, we already showed that plants differentially perceive MF variations in shoots and roots [35], with higher root oxidative stress responses [36], modulate ROS production and scavenging mechanisms [33], delay the transition to flowering, reduce leaf area index and flowering stem length and downregulate the flowering-related genes [34], alter clock gene amplitude [37], modulate gene expression pathways downstream of cryptochrome and phytochrome photoactivation [11], alter Fe and copper uptake efficiency and plant mineral nutrition [25,26] as well as lipid metabolism [38].

To assess whether the GMF impacts photosynthesis and the production of ROS, we evaluated the GMF influence by exposing a Lima bean to NNMF and GMF conditions. Our comparative analysis provides further evidence for the impact of the GMF on plant photosynthesis and oxidative status.

## 2. Results

### 2.1. The GMF Increases Leaf Area and Lowers Stomatal Density, but Does Not Alter the Water Content

In order to assess the effect of the GMF on Lima beans, we reduced the GMF intensity to NNMF conditions by using a triaxial Helmholtz coils system as previously detailed [34]. In this way, the direct comparison of NNMF and GMF data evidences the effect of the GMF in natural conditions at our latitude (see Materials and Methods for further details).

When exposed to NNMF, the Lima bean developed leaves with a lower area and a higher stomata density when compared to GMF plants (*p* < 0.05); however, the relative water content (RWC) was not significantly different between the NNMF and GMF treatments (Figure 1).

### 2.2. The GMF Modulates Chloroplast Morphology

After assessing the general morphology of leaves and stomata, we fixed the leaves of plants exposed to the GMF and NNMF and evaluated the ultrastructure and morphometry of the chloroplasts.

From the examination of several ultrathin sections, the dimensions of the chloroplasts were calculated by measuring the major and the minor axis. Figure 2 shows the density plot of the distribution of lengths indicating a general reduction in chloroplast dimensions in NNMF-exposed leaves.

Higher magnification of the chloroplasts allowed us to evaluate the morphometry of the chloroplast thylakoids (Figure 3). Plants exposed to the GMF showed a higher number of thylakoids per granum (*p* < 0.05) (Figure 3A,E). However, when the thickness of the granal thylakoids was measured, GMF-exposed plants showed a lower thickness (*p* < 0.05) (Figure 3B,E), when compared to NNMF-exposed plants (Figure 3G,H). Moreover, the thickness of intergranal thylakoids was lower in GMF-exposed plants (*p* < 0.05) (Figure 3C,E) when compared to NNMF plants (Figure 3G,H).

Finally, we evaluated the number of starch grains and plastoglobules per chloroplast in plants exposed to the GMF and NNMF. The number of both starch grains (Figure 4A,C) and plastoglobules (Figure 4B,C) was always higher (*p* < 0.05) in GMF plants when compared to NNMF (Figure 4D).

### 2.3. The GMF Impacts Leaf Total Protein, Carbon, Carbohydrate Content, δ^13^C and Chlorophyll Composition

Lima bean leaves exposed to the NNMF showed a reduction of total carbohydrates (*p* < 0.05) (Figure 5A), produced a higher amount of total proteins (*p* < 0.05) (Figure 5B) and a higher percentage of total carbon (*p* < 0.05) (Figure 5C). We then measured the carbon stable isotope composition (δ^13^C) and found that plants exposed to the NNMF had a more negative value than GMF-exposed plants (*p* < 0.05) (Figure 5D). To distinguish variations in the δ^13^C of the source CO_2_ from the effects of leaf metabolic processes, the δ^13^C signatures of leaf organic material (δ^13^C_plant_) were expressed as photosynthetic ^13^C discrimination, Δ, according to [39]. A higher photosynthetic ^13^C discrimination was observed in NNMF-exposed plants (21.06‰ ± 0.08) with respect to GMF plants (20.41‰ ± 0.06) (*p* < 0.05).

Chlorophylls and carotenoids were further investigated by the separation and analysis by HPLC-DAD. Two different types of chlorophyll were detected (Chlorophyll a, Chl a, and Chlorophyll b, Chl b) along with their respective degradation products, namely Chl a′ and Chl b′. Moreover, among the products derived from the degradation of chlorophylls, the uncolored Pheophytin a (Pheo a) and Pheophytin a′ (Pheo a′) were also detected. However, Pheophytin b, Pheophytin b′, and Pyropheophytin-related forms were not detected (<LOD). From a quantitative point of view, a reduction of both Chl a and Chl b was found in NNMF-exposed plants (*p* < 0.001) (Table 1). However, an increase of both Chl a′ and Chl b′ was found in NNMF-exposed plants (*p* < 0.001), which also showed an increased amount of Pheo a and Pheo a′ (*p* < 0.001), with respect to GMF-exposed plants (Table 1).

Exposure to either the GMF or NNMF did not affect the carotenoid profile from a qualitative point of view. The chromatographic conditions used for carotenoid analysis allowed the separation and quantification of two xanthophylls (lutein and a unknown putative xanthophyll) and seven carotenoids (Table 2). Regarding xanthophylls, plants exposed to the GMF showed an increased lutein content (*p* < 0.001), while no significant differences were found for the unidentified xanthophyll, with respect to the NNMF. The carotenoid content of *trans*-α-carotene, *cis*-α-carotene, *trans*-β-carotene and 9-*cis*-β-carotene was always higher in GMF-exposed plants (*p* < 0.001), whereas no significant differences were found for 15-*cis*-β-carotene, 13-*cis*-β-carotene and γ-carotene, with respect to NNMF-exposed plants (Table 2).

### 2.4. The GMF Impacts on RubisCO Content

Because the total content of carbohydrates was reduced in NNMF-exposed plants, the total protein extract was used to characterize the main enzyme involved in carbon assimilation, Ribulose-1,5-bisphosphate Carboxylase Oxygenase (RubisCO). We first analyzed the subunit composition by SDS capillary gel electrophoresis, which was performed by loading the same protein content of both GMF and NNMF Lima bean leaf protein extracts. Two clear bands related to the RubisCO large subunit (56 kDa), which is chloroplast-encoded, and the RubisCO small subunit (14 kDa), which is nuclear-encoded, are evident in both GMF- and NNMF-exposed leaves of Lima beans (Figure 6). A 26 kD band, most likely attributable to a light harvesting complex (LHCs) protein family, was also present in both the GMF and NNMF leaf extracts (Figure 6). Plants exposed to the GMF showed a higher density band than NNMF plants for both large and small RubisCO subunits as well as for the 26 kD band of LHCs (Figure 6).

In plants exposed to the GMF, the RubisCO activity was found to be 41.58 ± 2.80 nmol min^−1^ mg prot ^−1^, whereas in NNMF-exposed plants, the activity was lower (40.74 ± 3.01 nmol min^−1^ mg prot ^−1^). Despite the difference in the mean values, there was no statistical significance (*p* = 0.823) between the GMF- and NNMF-exposed plants for RubisCO enzyme activity.

### 2.5. The GMF Impacts on the Reaction Centers and the Reduction of Quinones

In dark-adapted Lima bean leaves of plants exposed to either GMF or NNMF conditions, the fluorescence kinetics was evaluated by OJIP analysis. The variable chlorophyll fluorescence (F_M_ − F_0_) = Fv showed no significant differences between the GMF and NNMF (Appendix A). In the NNMF, fluorescence intensity increased at 2 ms and 30 ms time points (*p* < 0.05), while at 300 ms it was not significantly (*p* = 0.065) higher, with respect to the GMF. We then measured S_m_ (the ratio between the area of the fluorescence induction curve and F_M_ − F_0_), a measure of the energy needed to close all reaction centers (RCs). In the term S_m_, the subscript “m” stands for multiple, referring to the multiple turn-over in the closure of the reaction centers. The more the electrons from the plastoquinone A (QA^−^) are transferred into the ET chain, the longer the fluorescence signals remain below F_M_ and the bigger S_m_ becomes. The NNMF plants always showed S_m_ values lower than the GMF (*p* < 0.05), indicating that every QA is reduced only once. In NNMF-exposed leaves, the turn-over number of QA, N, (calculated as the ratio between S_m_ (multiple turnover) and S_s_ (single turnover)), was lower than in the GMF plants (*p* < 0.05) (Figure 7A). N indicates how many times QA has been reduced in the time span from 0 to t_Fmax_. We also calculated the net rate of closure of the RCs (M_0_), the efficiency with which a trapped exciton can move an electron further than QA (Ψ_0_), the probability that an absorbed photon will move an electron into the ET chain and the flux of photons absorbed by the antenna pigments Chl* (ϕAbs). In NNMF-exposed plants, M_0_ was higher than in GMF-exposed plants (*p* < 0.05), whereas in GMF plants, Ψ_0_ and ϕAbs were always higher than in NNMF plants (*p* < 0.05). There was no significant difference in the maximum quantum yield of primary photochemistry (ϕPo) (≡TR_0_/ABS), whereas lower significant values were obtained in NNMF for ϕE_0_ (≡ET_0_/ABS), which represents the probability that an absorbed photon will move an electron into the ET chain. TR_0_/RC, which expresses the initial rate of the closure of RCs as a fractional expression over the total number of RCs that can be closed (resulting in the reduction of QA to QA^−^), was not significantly different between treatments, whereas in the NNMF, a higher flux of dissipated excitation energy at time zero per RC, DI_0_/RC, was present, with respect to the GMF (*p* < 0.05). The DI_0_/RC ratio increases due to the high dissipation of the inactive RCs (Figure 7B and Appendix A).

With regards the non-photochemical quenching (NPQ), NNMF-exposed plants showed a faster increase of NPQ upon exposure to actinic light (*p* < 0.05), even if the maximum value reached was similar for both the NNMF and GMF plants. No significant difference with regard to the GMF was found in the dark recovery phase (see also Appendix A).

A photosynthetic electron flow is coupled to the formation of a proton gradient across the thylakoid membrane. Such protons are then exploited by ATPase as a proton motif force (pmf) to produce ATP [40]. The pmf can be estimated by measuring the light dependent-electrochromic shift (ECSt) of carotenoid absorption [41]. The pmf was evaluated in the GMF and NNMF leaves upon exposure to different light intensities. The ECSt/chlorophyll measurements in the GMF and NNMF leaves showed no significant differences (Appendix A), despite the different morphologies of the thylakoid membranes in the two different samples (see Figure 3). Anyway, a tendency of increased pmf in NNMF leaves with respect to the GMF samples was observed at high light intensities (Appendix A).

To obtain insights into the organization of photosynthetic complexes, we performed a native Deriphat-PAGE analysis of the thylakoidal proteins. Thylakoid membranes were isolated from GMF and NNMF leaves and then solubilized with 0.7% α-dodecyl maltoside or 0.7% β-dodecyl maltoside. The Deriphat-PAGE (Figure 8), loaded on the basis of equal chlorophyll content, allowed the separation of several green bands according to their molecular size, with the smallest having the greatest electrophoretic mobility. These bands were referred to as free pigments, monomeric LHCII, trimeric LHCII, LHCII-CP29-CP24 supercomplex, PSII core, PSI-LHCI, and PSII supercomplexes, according to previous literature [42]. No major differences were evidenced in the profiles of pigment–protein complexes in the GMF and NNMF samples, suggesting that changes in the MF do not affect the assembly of photosynthetic supercomplexes on a chlorophyll basis.

SDS-PAGE analysis was then conducted to evaluate the protein accumulation in thylakoidal extracts from plants grown in the GMF or NNMF (Figure 9A). Coomassie-stained SDS-PAGE did not show major differences in the thylakoidal protein profiles. Both samples showed a dominant band attributed to LHCII polypeptides, surrounded by other light harvesting antenna polypeptides that abundantly accumulated in the SDS-PAGE region between 20 and 30 kDa. A protein band migrating at ~30 kDa can be attributed to the LHCB4 polypeptide [42,43]. Such antenna proteins appeared to be more accumulated in the NNMF sample on a chlorophyll basis (Figure 9A). Considering that such a polypeptide was shown to stoichiometrically accumulate with the PSII-core [42], it suggested that PSII was more abundant in the NNMF sample.

In order to verify whether the MF affects the assembly of photosynthetic supercomplexes on a chlorophyll basis, we conducted Western blot analyses using antibodies specific for representative proteins composing PSII and PSI (Figure 9B). In more detail, antibodies against PSBA (PSII core component) and LHCII (light harvesting protein complex) proteins were chosen to investigate PSII accumulation, whereas antibodies recognizing PSAA (PSI core component) and LHCA (PSI antenna system) polypeptides were used to quantify PSI accumulation. Western blot analyses did not provide significantly different accumulations of the polypeptides of interest. Anyway, a slight increase of PSBA, and thus PSII, and LHCII, and a partial decrease of PSI and LHCI were observed in NNMF plants compared to GMF plants.

Considering the reduced chlorophyll content measured in NNMF plants compared to GMF plants, the PSAA and LHCI content were further decreased in NNMF per dry weight (Figure 9C, D). All these considerations may be related to a possible perturbation of light absorption efficiency as observed in the above-discussed fluorescence analyses.

### 2.6. The GMF Modulates the MagR Homolog Genes IScA2 and CpIScA

We then assessed the gene expression of some Lima bean genes that show homology with the gene involved in magnetoreception, *MagR*, by focusing on the cytosolic *PlIScA1* and *PlIScA2* and the chloroplastic *PlcpIScA*.

When compared to the housekeeping gene *UBP6*, the expression of *PlIScA1* was unaffected (*p* = 0.063) by the reduction of the GMF to NNMF values, whereas a significant (*p* < 0.05) difference was found between the GMF and NNMF gene expression of both *PlIScA2* and *PlcpIScA* (Figure 10), where both genes were downregulated by the reduction of the MF.

### 2.7. The GMF Impacts on ROS Production and Gene Expression

The reduction of both chlorophyll and carotenoid content, the impairment in the photosynthetic apparatus and the downregulation of two *PlIScA* genes led to the assumption that the production of ROS could also be affected by the reduction of the MF. We then evaluated the production of ROS by assessing first the gene expression of superoxide dismutase (*SOD*), catalase (*CAT*), ascorbate peroxidase (*APX*), peroxidase (*PRX*), glutathione reductase (*GR*) and glutathione peroxidase (*GPX*), and then by quantifying the production of H_2_O_2_ and other peroxides.

qRT-PCR analyses show that in NNMF-exposed Lima bean plants, there was a higher downregulation of *CAT* and *PRX* with respect to GMF plants (*p* < 0.05) (Figure 8). *SOD* gene expression was not different between the GMF and NNMF (*p* > 0.05), whereas an upregulation was observed for *APX* in NNMF-exposed plants (*p* < 0.05). Of particular interest was the opposite regulation of *GR* and *GPX*, which were always downregulated in the GMF and upregulated in NNMF-exposed plants (Figure 11).

We then evaluated the production of H_2_O_2_ and other peroxides by using the MAK311 assay for H_2_O_2_ production and the PEROXsay assay for the total peroxides. Exposure of Lima beans to NNMF conditions significantly (*p* < 0.05) lowered both the total peroxides (Figure 12A) and H_2_O_2_ (Figure 12B) production.

## 3. Discussion

Changes in the MF, as with variations in temperature, light, water, and minerals, are perceived by plants as stress factors [35,44]. MF variations interfere with physiological processes, including photosynthesis [45], nutrient uptake [38], and modulation of genes involved in ROS and the production of polyphenolic antioxidants [33]. The results of this work indicate that GMF plays an important role in maintaining the oxidative status and proper photosynthetic activity. Moreover, we showed that chloroplast dimensions and thylakoid structures require the GMF, which is also determinant in the functioning of the light and dark phases of photosynthesis. Finally, we showed an interesting correlation between the regulation of some *IScA* and the photosynthetic activity.

Literature data report that the GMF alters plants’ gas exchange and metabolism [46], stomatal conductance and chlorophyll content [47,48]. In the mesophyll cells, the chloroplast structure is affected by GMF variations [49], while priming seeds with an increasing MF intensity alters the photosynthesis and growth parameters [50], Chl a, Chl b, carotenoids and total chlorophyll contents [51], and improves Photosystem II efficiency [52,53]. In addition, chlorophyll has Mg ions in its structure that can undergo MF action by electromagnetic interactions [54]. Interestingly, both Mg transport and uptake are altered by the MF [55] and may be correlated to the chlorophyll variations observed in this work. 

We noticed that both the stomatal density and leaf area were altered by reducing the GMF to NNMF values. Leaf δ^13^C may reflect a range of physiological responses, including stomatal conductance, altered C:N allocation to carboxylation and leaf structure [56]. δ^13^C_leaf_ is a reliable estimator of intrinsic leaf-level transpiration efficiency [57], and our results show that the increased number of stomata in NNMF-exposed plants correlates with a higher Δ value.

From a metabolic point of view, the increasing content of total carbohydrates in plants exposed to the GMF may have occurred due to the increase in photosynthetic efficiency. In this context, the biosynthesis of both chlorophylls and carotenoids plays a crucial role. The stimulating effects of an increasing MF on chlorophyll synthesis were also found by other authors in both algae and higher plants [54,58], while the photosynthetic machinery was enhanced by MF treatment, with particular reference to RCs [59]. The results of our investigation show an increased content of RubisCO in the GMF, but a similar enzyme activity in NNMF-exposed plants. Increasing the intensity of the GMF was shown to increase the soybean intensities of the bands corresponding to a larger subunit (53 KDa) and smaller subunit (14 KDa) of RubisCo [60], and the same results were observed in MF-primed seeds ([61] and reference therein). In *Spirulina platensis*, electrophoretic analyses of MF-exposed cells increased the large and small RubisCO subunits [54]. Increased RubisCO content, together with improved photosynthetic electron transport (Figure 7), suggest that GMF plants are potentially more efficient in using the light energy available to produce sugars through the photosynthetic process. Moreover, NNMF plants were characterized by a reduced PSI and LHCI content per dry weight: PSI being essential for the desaturation of photosynthetic electron transport, the improved photosynthetic electron flow evidenced in the GMF plants is likely related to the different PSI content. Accordingly, because NNMF plants are more prone to saturate the photosynthetic electron transport, NPQ activation was faster in these plants compared to GMF plants (Appendix A). The observation that the content of RubisCO increases but not its enzyme activity suggests that the increased carbohydrate content in GMF plants might depend on the higher quantity of available enzymes, independently of its activity.

Although the process through which the GMF affects living organisms is still far from being fully disentangled, the biocompass model based on the MagR/Cry complex has been demonstrated in the model insect *Drosophila melanogaster* [19]. Furthermore, it has recently been suggested that plant IScA proteins, homologous to the animal MagR, are good candidates for a better understanding of plant responses to changing MFs [24]. Indeed, among the different possible mechanisms of magnetoreception, at least two (the radical pair mechanism (RPM) of chemical magnetosensing and the MagR/Cry biocompass) adequately explain the alterations in the MF by the rates of redox reactions and subsequently altered concentrations of free radicals and ROS observed in different organisms [16,20,21,22]. Our results show that both the *PlIScA2* and *PlcpIScA* of Lima beans are modulated by the MF, with a significant downregulation in NNMF conditions corresponding to a reduction of electron flow from QA to the ET chain and a reduced efficiency to move electrons further than QA, with a consequent lower dissipation of interactive RCs. The cpIScA protein serves as a scaffold in chloroplast Fe- S cluster assembly and plays crucial roles in plastids, participating in photosynthesis and other metabolic pathways [62] with a higher expression level in green, photosynthetic tissues [63]. The loss of IScA causes severe defects in the accumulation of chloroplast Fe-S proteins, a dysfunction of photosynthesis, and a significant intracellular iron overload [64]. Interestingly, IScA could also act as a magnetic sensor by forming a magnetosensor (MagS) complex with Cry [65]. Higher plant photosystem II RC shares many features with bacterial RC, including a high-spin iron. In the RC of the purple bacterium *Rhodobacter sphaeroides* with reduced acceptor QA, the yield of triplet products is observed to be lowered by a weak external MF that decreases the population of the triplet states, due to the dependence of their energies on the magnitude of the field, which results in reduced singlet-triplet conversion [66,67]. We hypothesize that a similar mechanism might be associated to the loss of photosynthetic efficiency in NNMF-exposed plants.

Macromolecular aggregates of LHC have a considerable magnetic susceptibility, which enables the particles to rotate and align with their nematic axes parallel with the MF force, H. LHC, which are embedded in the thylakoids with a transmembrane direction, align perpendicular to H, and the relationship between the LHC content of various photosynthetic membranes and their capacity for alignment suggest that LHC might be the torque ordering chloroplasts in a MF [68]. Increasing the MF causes an enhancement of photosystem I fluorescence emission intensity, followed by a slow relaxation on the removal of the MF [69]. As MF is essentially equivalent to a spin perturbation, these effects can be explained in terms of spin re-organization, illustrating a memory effect via membrane re-alignment and assembly [69]. LHCII were reported to be involved in the organization of granal thylakoid membranes [70]: even if the LHCII content per chlorophyll was only slightly higher in NNMF vs. GMF plants, the different MF at which the plants were exposed may have caused a reorganization of LHCII in the thylakoid, thus altering the thickness of the grana. It is worth noting that even if the PSII content in NNMF plants was similar to the GMF plants case, the Chl a fluorescence induction curves kinetics suggests an increased dissipation of absorbed light energy due to inactive PSII RCs (DI_0_/RC), further supporting the stressing conditions at which NNMF plants were exposed by the reduction of the MF.

MFs with a higher intensity than the GMF have been reported to induce oxidative stress [71,72], suggesting the involvement of the radical pair mechanisms [73] with the contribution of chloroplasts [74]. During charge separation in photosynthetic RCs, O_2_ reacts with the triplet states and generates ^1^O_2_ [75]. The MF was also found to protect against photoinhibition of PSII, suggesting that radical pair recombination is responsible for a significant part of ^1^O_2_ production in the chloroplast [20]; a fast-relaxing high-spin ion generates an effective MF, thus suppressing triplet state formation as in the so-called radical pair mechanism [66]. Radical pair recombination contributes significantly to ^1^O_2_ production in the chloroplast, which in turn inhibits the repair of light-induced damage in photosystem II [20], consistently with the less efficient photosynthetic electron flow observed in the NNMF compared to GMF plants (Figure 7). Our results confirm previous findings that plants, under normal GMF conditions, modulate ROS scavenging enzymes and H_2_O_2_ production, indicating a functional role of plant magnetoperception in response to stress [33]. MF variations, which are an abiotic stress factor, may trigger signal transduction pathways that involve the plant alteration of the redox status.

## 4. Materials and Methods

### 4.1. Plant Materials and Growth Conditions

*Phaseolus lunatus* L. (cv Ferry Morse var Jackson Wonder Bush) seeds were obtained from the Max Planck Institute of Chemical Ecology in 2006 and multiplied in Italy. Seeds were soaked for 30 min in distilled water and then seeded in 10 cm pots containing a mixture of sterilized soil and vermiculite. Pots were immediately transferred under either the NNMF or GMF where they germinated and grew under 120 μmol m^−2^ s^−1^ light provided by a tunable LED lighting system source (PHYTOFY RL 150 W, Osram, München, DE) at 22 °C (±1.5 °C) with a 16/8 light/darkness photoperiod. For the entire experiment, the temperature was set and maintained by air conditioning. All experiments were performed under normal gravity and atmospheric pressure.

### 4.2. Near Null Magnetic Field (NNMF) Generation System and Plant Exposure

The local GMF values where typical of the Northern hemisphere at 45°0′59″ N and 7°36′58″ E coordinates. The near-null magnetic field (NNMF) was generated as previously described [34]. Real-time monitoring of the MF in the plant exposure chamber was achieved with a three-axis magnetic field sensor (model Mag-03, Bartington Instruments, Oxford, UK) that was placed at the geometric center of the Helmholtz coils. The output data from the magnetometer were uploaded to a VEE Pro software for Windows Release 7.51 (Agilent Technologies, https://www.keysight.com/it, accessed on 1 December 2022) to accurately adjust the current applied through each of the Helmholtz coil pairs in order to maintain the MF constant inside the plant growth chamber at NNMF intensity, as recently reported [35]. Pots containing *P. lunatus* were placed in the geometric center of the triaxial Helmholtz coils system and exposed to the NNMF. After an exposure period of four weeks, leaves were harvested and immediately frozen in liquid nitrogen. Appendix A shows the typical values of the MF in the triaxial Helmholtz coils with the presence of a current in the coils (i.e., NNMF) and without the current in the coils (i.e., GMF). As it can be seen, no other sources of MFs alter the values of the GMF.

### 4.3. Stomatal Density, Leaf Area and Relative Water Content (RWC)

Stomatal density was calculated by observing portions of the abaxial leaf surface directly with a stereomicroscope and the density calculated as the ratio between the stomata number and surface area. The leaf area was obtained from a leaf picture and by using the ImageJ 1.52a software (National Institute of Health, USA). The relative water content (RWC) percentage was calculated according to [76].

### 4.4. Transmission Electron Microscopy and Chloroplast Morphometry

Leaves of plants exposed to the NNMF and GMF were fixed by immersion in 3% (*v*/*v*) glutaraldehyde in a 100 mM sodium cacodylate buffer, pH 7.5, for 3 h under vacuum at room temperature. Samples were then washed in a sodium cacodylate buffer for 1 h and then pot-fixed in a 2% solution of osmium tetroxide in 50 mM sodium cacodylate buffer for 1 h at room temperature. Samples were then dehydrated with a graded series (30%, 50%, 70%, 95%, 100%) ethanol for 10 min for each step. In total, 100% ethanol was repeated three times. Samples were then transferred to propylene oxide and used as a transitional fluid before resin embedding. Samples were then embedded in an Epon–Araldite concentration of propylene oxide–resin mixture, and then polymerized for 24 h at 60 °C in pure Epon–Araldite resin. Ultrathin sections were then cut with a Reichert Ultracut ultramicrotome. Ultrathin sections were stained with lead citrate and uranyl acetate, mounted on 400 mesh grids and observed under the transmission electron microscope (TEM talos—Thermofisher) operating at 120 KV.

Images were acquired by a Ceta camera 4kx4k and the morphometry of Lima bean chloroplast in plants exposed to the GMF and NNMF was performed on ultramicrographs obtained by ultrasectioning different leaves.

### 4.5. RNA Preparation, cDNA Cloning and qRT-PCR Assays

Total RNA was isolated and purified from three independent samples of Lima bean leaves obtained from different plants (N = 12) by using TRIzol reagent (Thermo Fisher Scientific, Walthman, MA, USA). For RNA extraction, only secondary fully developed leaves were sampled. Extracted RNA quality and quantity were checked by using the nanospectrophotometer BioSpec-nano (Shimadzu, Kyoto, Japan) according to the manufacturer’s instructions. 

cDNA was synthesized from 500 ng of total RNA using random primers and the High-Capacity cDNA Reverse Transcription Kit (Applied Biosystems, Foster City, CA, USA), according to the manufacturer’s recommendations.

The obtained cDNA was diluted 1:5 and used for qRT-PCR assays. All experiments were performed on a QuantStudio 3 Real-Time PCR System (Applied Biosystems, Foster City, CA) using SYBR green I with ROX as an internal loading standard. The reaction was performed with a 25 µL mixture consisting of 12.5 µL 2X MaximaTM SYBR Green qPCR Master Mix (Thermo Fisher Scientific, Walthman, USA), 0.5 µL cDNA and 250 nM primers (Integrated DNA Technologies, Coralville, IA, USA). The controls included non-template controls (water template). The PCR conditions were the following for all primers: 10 min at 95 °C, 45 cycles of 30 s at 95 °C, 1 min at 60 °C, and 20 s at 72 °C. Fluorescence was read following each annealing and extension phase. All runs were followed by a melting curve analysis from 55 to 95 °C. All amplification plots were analyzed with the MX3000PTM software to obtain Ct values. *PlUBP6* and *PlActin1* were used as reference genes.

Ct values were analyzed using the Δ^Ct^ method, and the statistical difference between MF conditions were evaluated. Primers used for real-time PCR were designed using the Primer-BLAST (http://blast.ncbi.nlm.nih.gov/Blast.cgi, accessed on 1 December 2022) software and are shown in Appendix A.

### 4.6. Chlorophyll and Carotenoid Extraction

Chlorophylls, chlorophyll degradation products and carotenoids were extracted from Lima bean leaves exposed to either the GMF or NNMF according to the previously described protocol [77] with minor modifications [33]. After using liquid nitrogen to rapidly crush 100 mg of leaves, 1 mL of acetone was added to each sample to extract photosynthetic and non-photosynthetic pigments. To improve the extraction efficiency, the extraction process was repeated three times. The solid residue was separated from the supernatant after centrifugation (5000× *g*, 5 min, 4 °C), and the different acetone phases were pooled, collected and dried under a nitrogen flow before being dissolved in 6 mL of dimethylformamide (DMF). The DMF fraction was then centrifuged at 2000× *g* after being treated with 2 mL of hexane. The hexane from the upper layer containing carotenoids was separated and transferred to a new collection tube, while the DMF phase containing chlorophylls and their degradation products was treated with 6 mL of a 2% (*w*/*v*) aqueous solution of sodium hydroxide in an ice bath to remove potential polyphenol contamination. Then, 1 mL of a 1:1 (*v*/*v*) hexane:ethyl ether mixture was added and, after discarding the aqueous phase, the organic phase was completely dried under oxygen flow and reconstituted in a known volume of methanol (MetOH) and methyl tertiary butyl ether (MTBE) mixture in a 1:1 (*v*/*v*) ratio. Both the carotenoid and chlorophyll fractions were centrifuged at 10,000× *g*, at 4 °C for 10 min before injection into HPLC.

### 4.7. Liquid Chromatography of Chlorophylls, Chlorophyll Degradation Products and Carotenoids

A high-performance liquid chromatograph (1200 HPLC, Agilent Technologies, Santa Clara, CA, USA) was used to analyze samples grown under either GMF or NNMF condition. The molecules were separated, identified, and quantified as previously described [77]. The mobile phases (Solvent A [90% (*v*/*v*/*v*) MetOH, 3% (*v*/*v*/*v*) MTBE, and 5% (*v*/*v*/*v*) H_2_O]; Solvent B [88% (*v*/*v*/*v*) MTBE, 10% (*v*/*v*/*v*) MetOH, and 2% (*v*/*v*/*v*) H_2_O]) were fluxed in a thermally (25 °C) equilibrated C30 column (250 mm 2.1 mm i.d., 3 m, YMC America, Devens, MA, USA). In order to separate the pigments of interest, Solvent A and Solvent B were flushed at a constant flow rate of 0.2 mL/min following the ratio reported in Appendix A. The compounds eluted from the column at different retention times (RT) were detected by a diode array detector (DAD) set at the following wavelengths: 661 nm (Chl a, Chl a′), 642 nm (Chl b, Chl b′), 667 nm (Pheo 566 a, Pheo a′) and 460 nm (carotenoids). Pigment identification and quantification were performed based on injections of pure standards.

### 4.8. Total Sugar Content of Lima Bean Leaves

Total carbohydrates from *P. lunatus* leaves were measured using the Total Carbohydrate Assay Kit (MAK104, Sigma-Aldrich, St. Louis, MO, USA) according to the manufacturer’s instructions. Briefly, 50 mg of Lima bean leaves exposed to GMF or NNMF conditions were homogenized in 200 µL ice-cold Assay Buffer. Samples were then centrifugated at 13,000× *g* for 5 min and the supernatant was used for the assay. Samples (30 µL) were loaded to a flat bottom 96 well plate with the addition of 150 µL H_2_SO_4_. After 15 min at 90 °C in the dark, 30 µL of Developer was added to each well at 90 °C in the dark. The absorbance was read at 490 nm with a Microplate Reader (NB-12-0035, Neo Biotech, Nanterre, France). The amount of total carbohydrates present in the samples was determined from a standard curve made by dilutions of 2 mg mL^−1^ d-Glucose standard.

### 4.9. Total Protein Content of Lima Bean Leaves

The total protein content was obtained using an extraction buffer made of 62.5 mM Tris-HCl pH7, 10% Glycerol, 2% SDS, 1 mM EDTA and 1 mM PMSF. One hundred mg leaves and 40 mg TSSMs were grinded in liquid nitrogen. Samples were heated at 85 °C for 10 min and then centrifuged at 18,000× *g* for 5 min at room temperature (RT). The supernatant was then precipitated with 1:4 (*v*/*v*) 100% acetone, kept at −20 °C for 3 h and then centrifuged at 18,000× *g* for 7 min at RT. The supernatant was discarded, and the pellets were dried with airdrying for 20 min. Pellets were resuspended with a solution of 5 mM Tris-HCl pH 7, 1 mM EDTA and 0.1% SDS. Samples were then used for the protein electrophoresis and quantified by a Coomassie (Bradford) Protein Assay Kit (Thermo Fisher Scientific, Walthman, MA, USA).

### 4.10. Total Carbon and Carbon Isotope Discrimination (δ^13^C) Analyses

Samples were prepared by adding 1 mg of dry powdered leaves into 5 × 9 mm tin capsules. Capsules were carefully closed by folding them with cleaned tweezers and then transferred to an auto-sampler. Total carbon content and δ^13^C values of the samples were measured using a Flash 2000 HT elemental analyzer coupled, via a ConFLo IV Interface, with a Delta V Advantage isotope ratio mass spectrometer (IRMS), interconnected to the software Isodat 3.0 (Thermo), according to Bononi and co-workers [78]. Briefly, the combustion/reducing reactors, combined in a single quartz tube, were heated at 1020 °C. The He gas flow was 120 mL min ^−1^ and 100 mL min ^−1^ for the carrier and reference, respectively. The O_2_ purge for flash combustion was 3 s at a flow rate of 175 mL min ^−1^ per sample. The GC separation column was maintained at 45 °C. Total carbon was determined on combusted gases after GC separation by a thermal conductivity detector. Calibration was performed using solid in-house standards. For δ^13^C determination, the CO_2_ reference gas pulse was introduced three times (20 s each) at the beginning of each run. The run time of the analysis was 600 s for a single run. The analysis of each sample was performed five times. Calibration was performed using three secondary reference materials provided by IAEA: NBS18 (δ^13^C = −5.014 ± 0.035‰); IAEA-600 (δ^13^C = −27.771 ± 0.043‰); and IAEA-612 (δ ^13^C = −36.722 ± 0.006‰). Two solid in-house standards, sulfanilamide (δ^13^C = −27.23 ± 0.06‰) and methionine (δ^13^C = −30.01 ± 0.05‰), were used for normalization and quality assurance. The isotope ratio ^13^C/^12^C was expressed using the standard δ^13^C notation: δ^13^C = [(^13^C/^12^C) sample/(^13^C/^12^C)VPDR ^−1^] × 1000, which expresses the part per thousand deviation of the isotope ratio ^13^C/^12^C of a sample relative to an international standard, the Vienna Pee Dee Belemnite (VPDR; [79]). The photosynthetic ^13^C discrimination was calculated according to [39] as reported in Equation (1).
(1)$$\Delta =\frac{\delta^{13}C_{atm}-\delta^{13}C_{leaf}}{1+\delta^{13}C_{leaf}}$$

### 4.11. Capillary Gel Electrophoresis

Lima bean leaf proteins from plants exposed to GMF and NNMF were characterized with the Agilent Protein 80 Kit by using the Agilent 2100 Bioanalyzer (Agilent Technologies, Santa Clara, CA, USA) according to the manufacturer’s instruction. The same concentration of proteins for each sample obtained from the total protein extraction was used to prepare samples, reaching a final volume of 4 µL. The Protein 80 Chip was then analyzed in the 2100 Bioanalyzer, and the data acquisition and analysis were performed by the Agilent 2100 Expert Software (Agilent Technologies, Santa Clara, CA, USA).

### 4.12. RubisCO Extraction and Enzyme Activity

Enzyme extraction and purification were conducted at 4 °C. A ratio of 1:6 (*w*/*v*) cold extraction buffer (50 mM NaOH-Bicine pH 8.2, 20 mM MgCl_2_, 1 mM EDTA, 2 mM Benzamidine, 5 mM aminocaproic acid, 50 mM 2-mercaptoethanol, 10 mM DL-dithiothreitol (DTT), 1 mM phenylmethylsulfonyl fluoride (PMSF)) was added to samples of 40 mg (1:12.5 *w*/*v*) of GMF- or NNMF-exposed Lima beans that were previously grinded in liquid nitrogen. The homogenate was then centrifuged at 14,000× *g* for 5 min at 4 °C. The supernatant was transferred to a new ice-cold tube and immediately used for the RuBisCO activity assay.

The assay was carried out in a Microplate reader (NB-12-0035 Neo Biotech, Nanterre, FR). In flat bottom 96-well polystyrene microplates, 159.9 µL of mQ water for the blank and 153.9 µL for the samples were pipetted into each well, followed by a 35.1 µL assay mix (100 mM NaOH-Bicine pH 8.2, 20 mM MgCl_2_, 10 mM NaHCO_3_, 20 mM KCl, 5 mM DTT, 2 UI 3-Phosphoglyceric phosphokinase, 0.4 UI α-Glycerophosphate Dehydrogenase, 24 UI Triosephosphate isomerase, 2.8 UI Glyceraldehyde 3-Phosphate Dehydrogenase, 3 mM ATP, 1 mM NADH) avoiding exposure to light. Next, 5 µL of a sample supernatant were added to the wells. Then, 6 µL 20 mM RuBP (ribulose-1,5-bisphosphate) were added to start RubisCO activity. Then, 5 µL of the sample supernatant were added to the wells. The Microplate reader was set at 30 °C and absorbance was read at 340 nm. The calculation of RubisCO activity was carried out according to Sales and co-workers [80].

### 4.13. H_2_O_2_ Quantification

Determination of the H_2_O_2_ content was carried out by using a MAK311 Peroxide Assay Kit (Sigma-Aldrich, St. Louis, MO, USA). First, 50 mg of GMF- or NNMF-exposed leaves were grinded in liquid nitrogen and extracted in 1:10 (*w*/*v*) mQ water. After centrifugation at 15,000× *g* for 10 min, the supernatant was used for the assay. In a flat bottom 96-well plate, 200 µL of detection reagent were added to each 40 µL sample. After 30 min incubation, the absorbance was read at 600 nm at the microplate reader. The H_2_O_2_ content was then calculated from a standard curve made by dilutions of 3% H_2_O_2_ standard.

### 4.14. Determination of Total Peroxides Content

The quantitative content of peroxides in the samples was measured employing a PEROXsay Assay Kit (G-Biosciences, St. Louis, MO, USA). Overall, 150 mg of GMF- or NNMF-exposed leaves were homogenized with a micropestle in liquid nitrogen and 1 mL of mQ water was added for peroxides extraction. Samples were then centrifuged at 15,000× *g* for 10 min at 4 °C. The supernatant was used for the assay. In a flat bottom 96-well plate, to each sample, 10 volumes of assay solution were added, mixed and incubated at room temperature for 30 min. After incubation, absorbance was read at 570 nm through the use of a microplate reader. The amount of total peroxides present in the samples was determined from a standard curve made by dilutions of 30% hydrogen peroxide solution.

### 4.15. Chl a Fluorescence Kinetics

Chlorophyll fluorescence parameters Ft, QY, NPQ, OJIP, and Light Curve (QY) were obtained with a FP100 fluorometer (Photon Systems Instruments, Drásov, Czech Republic). Measurements were performed by following the manufacturer instructions.

### 4.16. Membrane Preparation, Gel Electrophoresis and Immunoblotting

Thylakoid membranes were isolated as previously described [81]. SDS-PAGE analysis was performed using the Tris-Tricine buffer system [82]. Non-denaturing Deriphat-PAGE was performed following the method developed in [83].

### 4.17. Electrochromic Shift Measurement

Proton motive force upon exposure to different light intensities was measured by electrochromic shift (ECS) with MultispeQ V2.0 (PhotosynQ) as detailed in [84].

### 4.18. Statistical Analysis

The data obtained were statistically treated by using Systat 10. Mean value was calculated along with the SD. Paired t-test and Bonferroni adjusted probability were used to assess the difference between treatments and controls. At least three biological replicates were always performed.

## 5. Conclusions

Here, we provided evidence that the GMF is required for the efficiency of the photosynthetic machinery in Lima beans. In our experimental approach, to demonstrate the role of the GMF, we used a system able to reduce the GMF to NNMF values and carried out a comparative analysis. As depicted in the general scheme of Figure 13, the GMF is involved in both structural and metabolic processes involved in Lima bean photosynthesis, from leaf structure (e.g., stomata and leaf area), to chloroplast morphology and thylakoid structure up to the metabolism of the major pigments involved in the light harvesting complexes. In GMF conditions, electrons move from QA to the ET chain and beyond, while the number of times that the QA is reduced increases, with respect to the NNMF. The GMF also favors the probability that a photon may move an electron into the ET chain by favoring a higher dissipation of interactive RCs. We found an interesting correlation between *Pl*cpIScA, the homolog of the magnetoreceptor gene *MagR*, and the photosynthetic efficiency, suggesting a role for this IScA in the modulation of MF-dependent responses in photosynthesis. *PLcpIScA* modulation was also correlated to the increased production of ROS, including H_2_O_2_ and other peroxides, confirming the tight relationship between magnetoreception and oxidative stress.

## Figures and Tables

**Figure 1 ijms-24-02896-f001:**
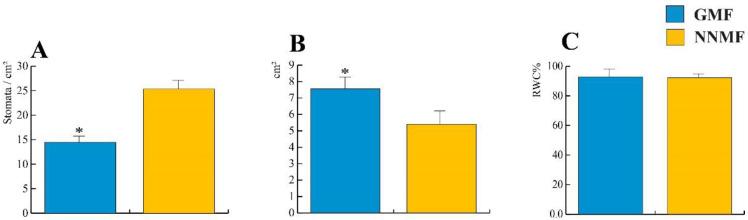
Effect of reducing the GMF on Lima bean leaves. (**A**) Stomatal density; (**B**) leaf area; and (**C**) relative water content (RWC) %. Bars represent standard deviation, asterisk indicates significant (*p* < 0.05) differences between GMF and NNMF.

**Figure 2 ijms-24-02896-f002:**
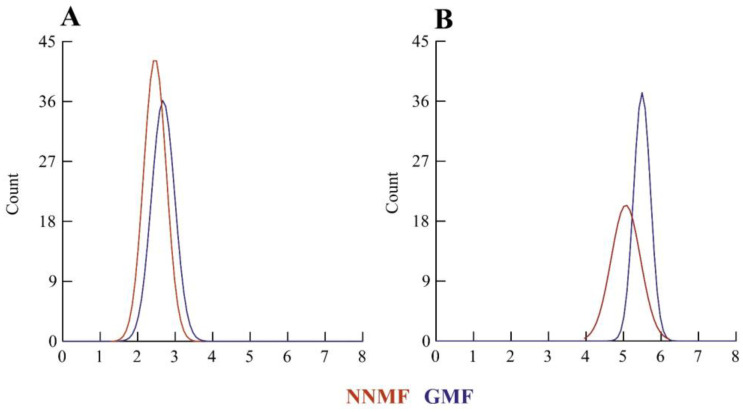
Morphometry of Lima bean chloroplasts in plants exposed to the GMF and NNMF. Density plots of minor axis length (**A**) and major axis length (**B**). X-axis values are expressed as microns.

**Figure 3 ijms-24-02896-f003:**
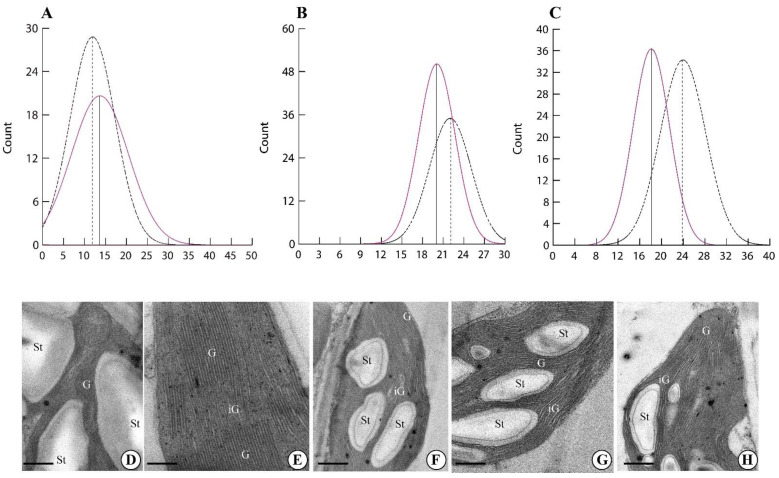
Morphometry of Lima bean chloroplasts in plants exposed to the GMF and NNMF. Density plots of the number of thylakoids per granum (**A**), granal thylakoid thickness expressed in nm (**B**) and intergranal thylakoid thickness expressed in nm (**C**). Dotted line, NNMF. Colored line, GMF. Ultrastructure of Lima bean chloroplasts in plants exposed to the GMF (**D**,**E**) and NNMF (**F**–**H**). G, granal thylakoids; iG, intergranal thylakoids; St, starch grain. Metric bars: (**D**), 1.50 µm; (**E**), 0.44 µm; (**F**), 1.20 µm; (**G**), 2.40 µm; (**H**), 2.40 µm.

**Figure 4 ijms-24-02896-f004:**
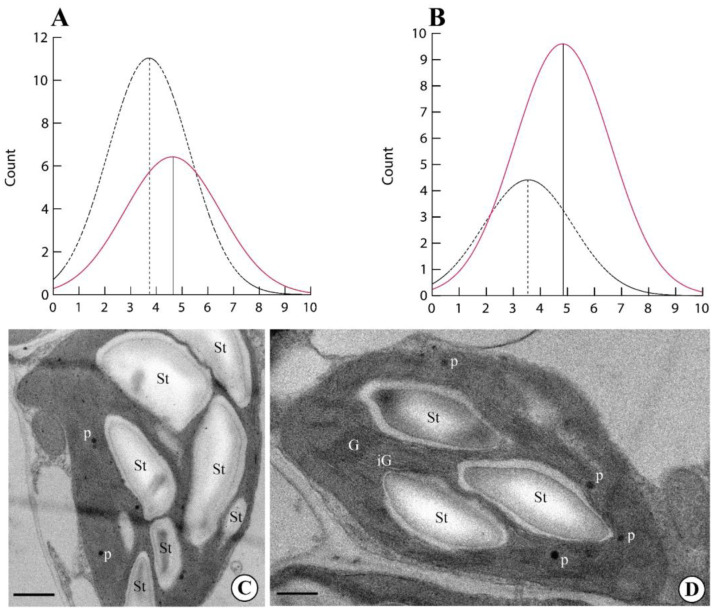
Morphometry of Lima bean chloroplasts in plants exposed to the GMF and NNMF. Density plots of number of starch grains per chloroplast (**A**) and number of plastoglobules per chloroplast (**B**). Dotted line, NNMF. Colored line, GMF. Ultrastructure of Lima bean chloroplasts in plants exposed to the GMF (**C**) and NNMF (**D**). G, granal thylakoids; iG, intergranal thylakoids; St, starch grain; p, plastoglobules. Metric bars: (**C**), 2.80 µm; (**D**), 3.80 µm.

**Figure 5 ijms-24-02896-f005:**
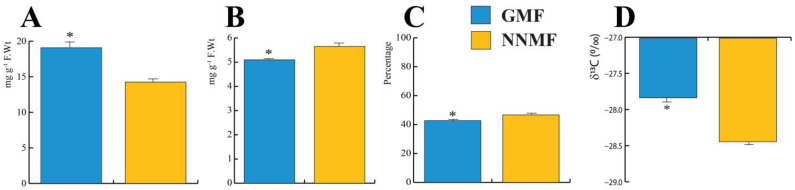
Total carbohydrate, protein, carbon content and δ^13^C of Lima bean plants exposed to the GMF and NNMF. (**A**) Total carbohydrate content; (**B**) total protein content; (**C**) percentage of leaf carbon; and carbon isotope discrimination δ^13^C (**D**). Bars indicate standard deviation, asterisk indicates significant (*p* < 0.05) differences between GMF and NNMF.

**Figure 6 ijms-24-02896-f006:**
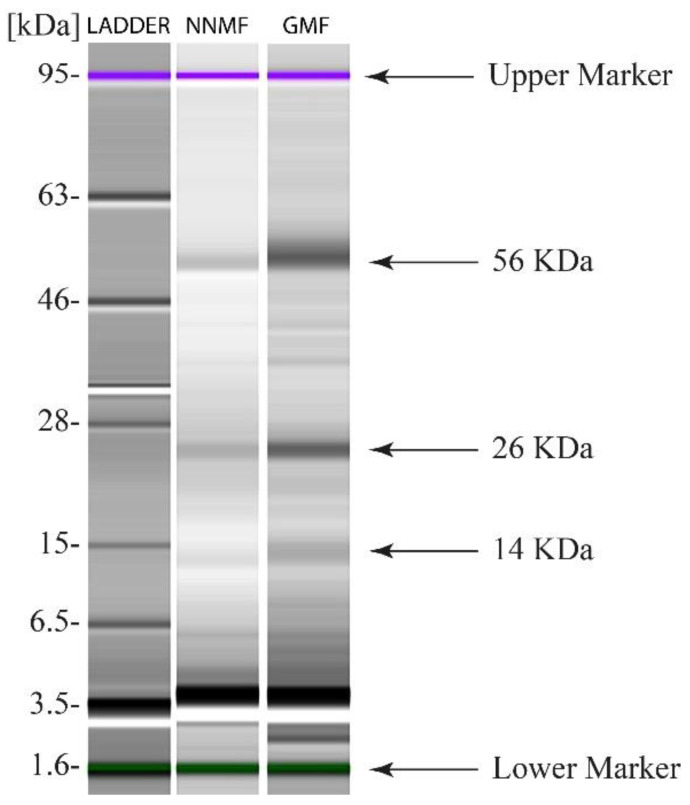
SDS Capillary gel electrophoresis (Agilent 2100 Bioanalyzer) of leaf proteins from Lima bean plants exposed to GMF and NNMF.

**Figure 7 ijms-24-02896-f007:**
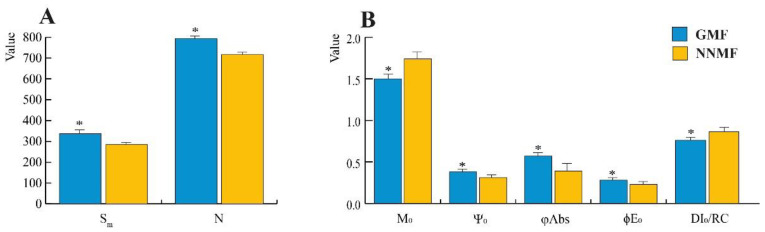
(**A**,**B**) Reaction centers, turnover number and quantum efficiency from the OJIP fluorescence analysis of Lima bean leaves exposed to either the GMF or NNMF. Asterisks indicate significant differences between the GMF and NNMF values. Metric bars indicate standard deviation. See text for symbol explanations.

**Figure 8 ijms-24-02896-f008:**
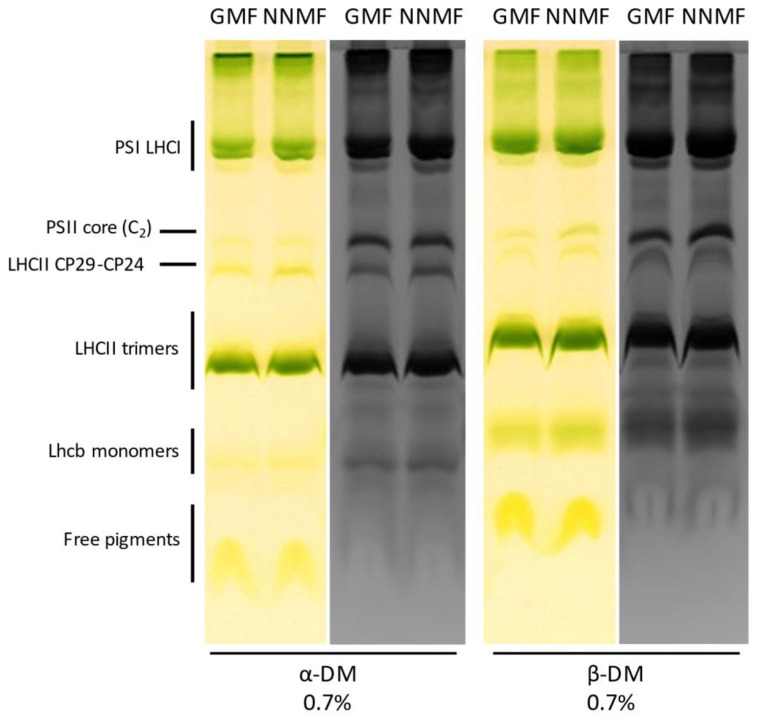
Analysis of membrane protein complexes in the GMF and NNMF thylakoids by Deriphat-PAGE (colored lanes), later stained with Coomassie (black and white lanes). Thylakoids have been solubilized with α-dodecyl maltoside or β-dodecyl maltoside before loading. Overall, 30 µg Chl were loaded per lane.

**Figure 9 ijms-24-02896-f009:**
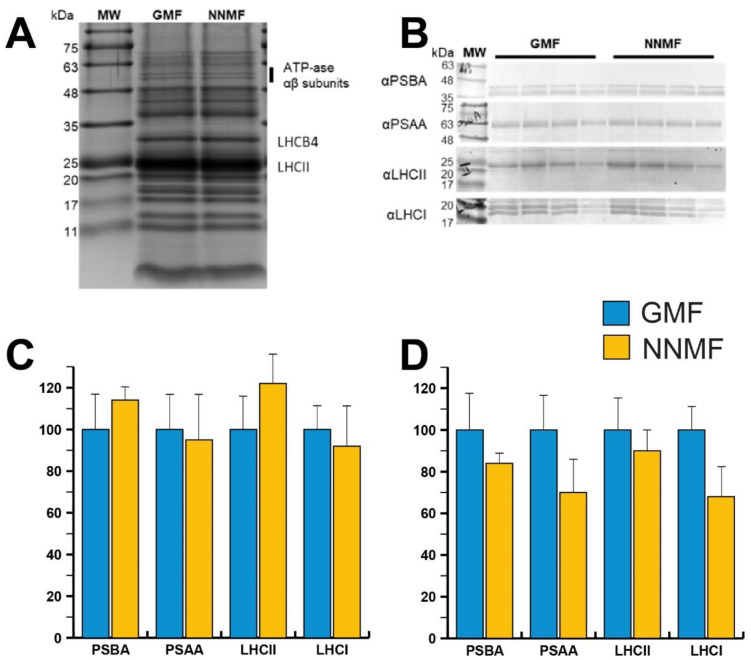
(**A**) SDS-PAGE analysis of protein complexes accumulated in GMF and NNMF thylakoidal membranes. Three micrograms of chlorophyll were loaded in each lane. (**B**) Immunoblotting used for the quantification of representative proteins (PSBA, PSAA, LHCII and LHCA) composing PSII or PSI in GMF and NNMF thylakoids. Loading corresponded to 2, 1.5, 1 and 0.5 µg of chlorophyll. (**C**) Relative accumulation of representative proteins indicated in (**B**), having protein accumulation in GMF samples as a reference (100%). (**D**) Relative accumulation per dry weight of representative proteins indicated in (**B**) calculated from chlorophyll content per dry weight, having protein accumulation in GMF samples as a reference (100%).

**Figure 10 ijms-24-02896-f010:**
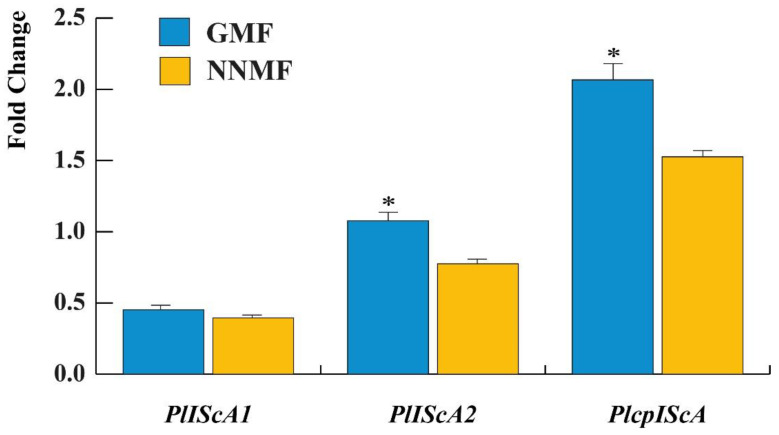
Gene expression of some *MagR* gene homologs in Lima bean: *PlIScA1*, *PlIScA2* and *PlcpIScA*. Data are expressed as a fold change with respect to the housekeeping gene *UBP6*. Metric bars indicate standard deviation, asterisk indicates significant (*p* < 0.05) differences between GMF and NNMF.

**Figure 11 ijms-24-02896-f011:**
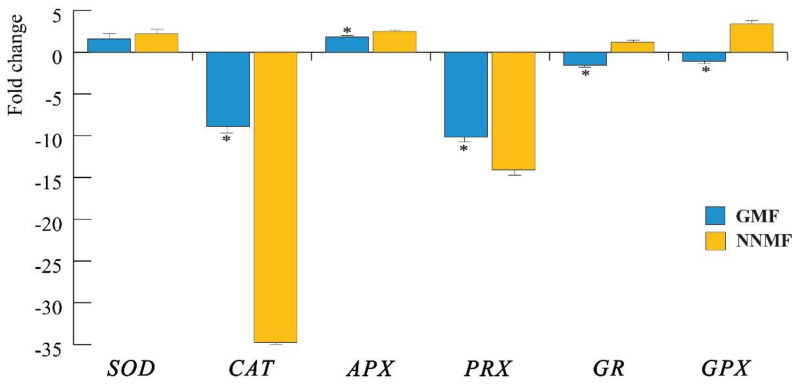
Expression of the main genes coding for enzymes involved in ROS metabolism. *SOD*, superoxide dismutase; *CAT*, catalase; *APX*, ascorbate peroxidase; *PRX*, peroxidase; *GR*, glutathione reductase and *GPX*, glutathione peroxidase. Data are expressed as fold changes with respect to the housekeeping gene actin. In order to emphasize the visualization of data, fold change values below zero were plotted as −1/value, in order to obtain negative fold change values (indicating downregulation). Metric bars indicate standard deviation, asterisk indicates significant (*p* < 0.05) differences between the GMF and NNMF.

**Figure 12 ijms-24-02896-f012:**
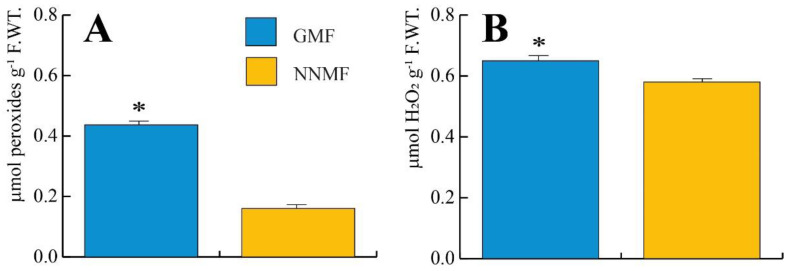
Production of total peroxides (**A**) and H_2_O_2_ (**B**) in Lima bean plants exposed to GMF and NNMF conditions. Metric bars indicate standard deviation, asterisk indicates significant (*p* < 0.05) differences between GMF and NNMF.

**Figure 13 ijms-24-02896-f013:**
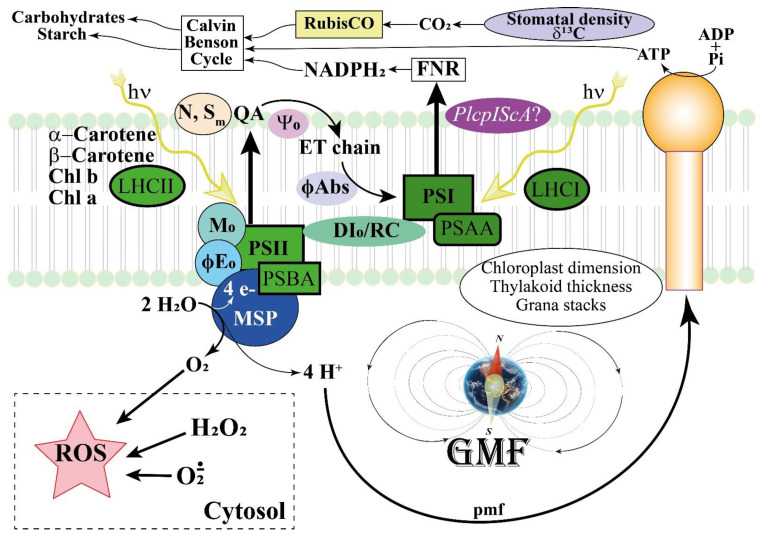
General scheme depicting the interaction between the GMF and the photosynthetic responses in Lima beans. The GMF regulates leaf area, stomatal density, and δ^13^C, which in turn influence the CO_2_ availability for RubisCO and the production of carbohydrates and chloroplast starch. Chloroplast dimension and thylakoid structure are also regulated by the GMF to maintain the correct number of grana stacks and thylakoid thickness. The content of photosynthetic pigments including Chl a and Chl b, the main carotenoids and the light harvesting complexes LHCI and LHCII are regulated by the GMF, whereas the pmf is less affected by MF variations. On the other hand, a regulation occurs for PSAA and PSBA, which are associated with Photosystem I and II, respectively. In GMF conditions, electrons from the plastoquinone A (QA) are preferentially transferred to the ET chain (S_m_) and the number of times that the QA is reduced increases (N). The GMF favors the efficiency to move electrons further than QA (Ψ_0_), with a higher probability that a photon may move an electron into the ET chain (ϕE_0_ ≡ ET_0_ /ABS) and to directly move electrons in the ET chain (ϕAbs). Moreover, the GMF favors a higher dissipation of interactive reactions centers (DI_0_/RC). The GMF upregulates the expression of Iron Sulfur Complex Assembly homologs of the magnetoreceptor gene *MagR*, with particular reference to *PlcpIScA*. This activity is correlated to the increased production of ROS, including H_2_O_2_ and other peroxides, which depend on the modulation of ROS producing and ROS scavenging genes.

**Table 1 ijms-24-02896-t001:** Chemical characterization of Lima bean chlorophylls in plants exposed to GMF and NNMF conditions. Values are expressed as µg g^−1^ F.Wt (standard deviation). *p* value refers to differences in chlorophyll content between GMF- and NNMF-exposed plants.

Compound	RT	GMF	NNMF	*p* Value
Chlorophyll a	15.2	37.62 (1.65)	25.83 (0.12)	<0.001
Chlorophyll b	17.2	9.52 (0.52)	7.31 (0.89)	<0.001
Chlorophyll a′	15.8	0.55 (0.02)	1.79 (0.05)	<0.001
Chlorophyll b′	17.7	0.11 (0.01)	0.22 (0.01)	<0.001
Pheophytin a	22.7	0.64 (0.05)	1.35 (0.02)	<0.001
Pheophytin a′	22.3	0.03 (0.01)	0.03 (0.01)	>0.050

**Table 2 ijms-24-02896-t002:** Chemical characterization of Lima bean carotenoids in plants exposed to GMF and NNMF conditions. Values are expressed as µg g^−1^ F.Wt (standard deviation). *p* value refers to differences in carotenoid content between GMF- and NNMF-exposed plants.

Compound	RT	GMF	NNMF	*p* Value
Lutein	15.3	5.12 (0.05)	3.15 (0.12)	<0.001
Putative xanthophyll	18.6	5.12 (0.16)	5.23 (0.09)	>0.05
15-*cis*-β-carotene	20.8	3.2 (0.15)	3.51 (0.04)	>0.05
13-*cis*-β-carotene	21.5	7.63 (0.14)	7.72 (0.25)	>0.05
*Trans*-α-carotene	21.8	7.85 (0.22)	5.78 (0.19)	<0.001
*cis*-α-carotene	22.6	19.68 (0.17)	4.54 (0.05)	<0.001
*Trans*-β-carotene	23.1	108.84 (1.4)	66.06 (1.97)	<0.001
9-*cis*-β-carotene	23.5	19.21 (0.41)	12.42 (0.39)	<0.001
γ-carotene	24.2	4.85 (0.07)	4.21 (0.13)	>0.05

## Data Availability

Data are available on request.

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
