# Peer review of "The Geomagnetic Field (GMF) Is Required for Lima Bean Photosynthesis and Reactive Oxygen Species Production"

_ijms, 2023, doi:10.3390/ijms24032896_

Round 1
Reviewer 1 Report
The manuscript ‘IJMS-2166914’ deals with very interesting and important topic of magnetic field influence on photosynthesis and ROS production in Lima bean plants. The results are valuable and good style of presentation gave me an opportunity for the conditionally acceptance of this manuscript with a subject to minor revision indicated below.
Minor notes/corrections;
(1) Figure 3 and Figure 4 have to be improved. This is very hard to see dashed lines for NNMF in these Figures. For the comparison, dashed lined are perfectly present in Figure 2. Please do similar in Figures 3 and 4.
(2) L165 and L557. High-performance liquid chromatography (HPLC) was described in M&M section but HPLC-DAD (HPLC with photodiode-array detection) was mentioned in L165 but not described in M&M section. Please manage it.
(3) L168-169 and Table 1. Please correct the spelling and use it identical ‘Pheophytin a’ as in Table 1 and in L169 for ‘b’ but not ‘Phaeophytin a’ as in L168.
(4) L217, L664, L717 and Suppl. Figure S1. Authors must be very careful with parameters of ‘Chlorophyll a fluorescence induction (OJIP)’. In the commonly used abbreviation (OJIP), the first position is Capital letter ‘O’ but not number ‘Zero’ (0). Letter ‘O’ and number ‘0’ looks very similar but meaning is very different. Only capitalized letter ‘O’ must be used in the name of this method. Please correct.
(5) L471. Please add brief information, what kind of genotype, cultivar or accession of Lima bean (Phaseolus lunatus L.) was used in this study? Where seeds were received from? Please do NOT write ‘From local market…’. It has to be more scientific and reproducible information for readers who wish to repeat such work.
(6) L471-472. Please clarify that pots with soaked seeds were placed in NNMF or GMF conditions. This means that seeds were germinated and plants were grown in such conditions from the beginning. This is important to note because it is possible to germinate seeds and move pots with seedlings to NNMF or GMF conditions.
(7) L489-490. Authors wrote: “After the exposure period…”. What was this ‘exposure period? One second? One minute? One hour? One day? One month? One year? Please clarify. Please also add important information, what stage of plant development was used for samples collecting and carry out all experiments?
(8) L515. “…three independent sampling of Lima bean leaves…”. What does this mean? Three leaves from one plant or one leaf from each of three plants or another? Please clarify. This is also important which leaf was collected for RNA extraction? Youngest, fully developed, in the middle of plant stem or old leaf from the bottom of plant?
(9) L516-517. “Sample quality and quantity was checked…”. This phrase is confusing particular after leaf sampling. Please modify that “Extracted RNA quality and quantity was checked…”.
(10) L519. The following phrase is absurd and nonsense: “…total RNA was retro-transcribed into cDNA…”. What does mean ‘retro-transcribed’? I know only ‘retrotransposon’ but this is not related to this term. I suppose authors want to say ‘reverse-transcribed’. This is acceptable. Alternatively, it is also possible to say ‘cDNA was synthesized from RNA…’. Please make corrections.
(11) L729. Authors still need to make their decision. Their presented research was funded or not? In current form, this is just a template from IJMS but Authors must be more careful with their manuscript submission.
(12) My last point is not compulsory and maybe more ‘philosophical’. Authors used Near Null Magnetic Field (NNMF) generation system, which reduced MF, but what is about another instrument or generation system, which can increase MF? Does this exist? If authors know something about it, could you please insert some information either in Introduction or in Discussion section. Just it would be interesting for readers to know, what is happened with Lima bean (or other plant species) in conditions with higher MF? If not, authors still can insert their statement like those: ‘Cases with increased MF and their influence on plants are unknown…”. Maybe, this is my imagination for your future research… Unfortunately, due to my recommendation as ‘Minor revision’, I cannot see your responses. However, I can find it later after the revised manuscript acceptance.
Author Response
The manuscript ‘IJMS-2166914’ deals with very interesting and important topic of magnetic field influence on photosynthesis and ROS production in Lima bean plants. The results are valuable and good style of presentation gave me an opportunity for the conditionally acceptance of this manuscript with a subject to minor revision indicated below.
R: We thank very much the reviewer for the appreciation of our work
Minor notes/corrections;
(1) Figure 3 and Figure 4 have to be improved. This is very hard to see dashed lines for NNMF in these Figures. For the comparison, dashed lined are perfectly present in Figure 2. Please do similar in Figures 3 and 4.
R: Figures 2-4 have been redrawn in colors to better evidence the drawings
(2) L165 and L557. High-performance liquid chromatography (HPLC) was described in M&M section but HPLC-DAD (HPLC with photodiode-array detection) was mentioned in L165 but not described in M&M section. Please manage it.
R: we thank the reviewer for noticing this. We added information on DAD in the Material and Methods section.
(3) L168-169 and Table 1. Please correct the spelling and use it identical ‘Pheophytin a’ as in Table 1 and in L169 for ‘b’ but not ‘Phaeophytin a’ as in L168.
R: we thank the reviewer for finding these typos that have been corrected.
(4) L217, L664, L717 and Suppl. Figure S1. Authors must be very careful with parameters of ‘Chlorophyll a fluorescence induction (OJIP)’. In the commonly used abbreviation (OJIP), the first position is Capital letter ‘O’ but not number ‘Zero’ (0). Letter ‘O’ and number ‘0’ looks very similar but meaning is very different. Only capitalized letter ‘O’ must be used in the name of this method. Please correct.
R: we thank the reviewer for noting this typo, which has been corrected throughout the text and Supplementary material
(5) L471. Please add brief information, what kind of genotype, cultivar or accession of Lima bean (Phaseolus lunatus L.) was used in this study? Where seeds were received from? Please do NOT write ‘From local market…’. It has to be more scientific and reproducible information for readers who wish to repeat such work.
R: The information on the variety of Lima Bean and the source (Max Planck Institute) has been added
(6) L471-472. Please clarify that pots with soaked seeds were placed in NNMF or GMF conditions. This means that seeds were germinated and plants were grown in such conditions from the beginning. This is important to note because it is possible to germinate seeds and move pots with seedlings to NNMF or GMF conditions.
R: a new sentence specifies that the seed were germinating and the plant developing from the beginning of the experiment in both NNMF and GMF conditions.
(7) L489-490. Authors wrote: “After the exposure period…”. What was this ‘exposure period? One second? One minute? One hour? One day? One month? One year? Please clarify. Please also add important information, what stage of plant development was used for samples collecting and carry out all experiments?
R: We thank the reviewer for noticing this important point. Information about the timing of exposure period has been added.
(8) L515. “…three independent sampling of Lima bean leaves…”. What does this mean? Three leaves from one plant or one leaf from each of three plants or another? Please clarify. This is also important which leaf was collected for RNA extraction? Youngest, fully developed, in the middle of plant stem or old leaf from the bottom of plant?
R: we added the information requested: “obtained from different plants (N = 12)”
(9) L516-517. “Sample quality and quantity was checked…”. This phrase is confusing particular after leaf sampling. Please modify that “Extracted RNA quality and quantity was checked…”.
R: The sentence has been rephrased by following the reviewer’s suggestion
(10) L519. The following phrase is absurd and nonsense: “…total RNA was retro-transcribed into cDNA…”. What does mean ‘retro-transcribed’? I know only ‘retrotransposon’ but this is not related to this term. I suppose authors want to say ‘reverse-transcribed’. This is acceptable. Alternatively, it is also possible to say ‘cDNA was synthesized from RNA…’. Please make corrections.
R: The sentence has been rephrased by following the reviewer’s suggestion
(11) L729. Authors still need to make their decision. Their presented research was funded or not? In current form, this is just a template from IJMS but Authors must be more careful with their manuscript submission.
R: We thank the reviewer for noticing this. We deleted the template sentence and left the right one
(12) My last point is not compulsory and maybe more ‘philosophical’. Authors used Near Null Magnetic Field (NNMF) generation system, which reduced MF, but what is about another instrument or generation system, which can increase MF? Does this exist? If authors know something about it, could you please insert some information either in Introduction or in Discussion section. Just it would be interesting for readers to know, what is happened with Lima bean (or other plant species) in conditions with higher MF? If not, authors still can insert their statement like those: ‘Cases with increased MF and their influence on plants are unknown…”. Maybe, this is my imagination for your future research… Unfortunately, due to my recommendation as ‘Minor revision’, I cannot see your responses. However, I can find it later after the revised manuscript acceptance.
R: We thank the reviewer for this comment. Our system is based on years of studies and development of a condition where plants under NNMF or GMF are continuously monitored to evaluate the fluctuations in the B value. We build a laboratory that completely lacks any instrumentation that may generate magnetic fields. Because of the importance of this issue we added a supplementary files that shows the typical variations inside the triaxial Helmholtz coils with current (NNMF) and without current (GMF) to show that no other sources of MF are present, but the GMF. A new sentence and a Supplementary Figure S4 in the Materials and Methods explains this condition.
Reviewer 2 Report
The manuscript by Parmagnani et al. considers the influence of geomagnetic field and near null magnetic field on photosynthetic apparatus, chloroplast structure, antioxidant proteins, production of peroxides and other. The work performed on high technical level and seems to be very interesting. However, I have some minor remarks:
(1) Please extend technical description of near null magnetic field generation system.
(2) Please add age of plants which were used in experiments.
(3) In figure 13, please add direction of changes which were induced by absence of magnetic field.
Author Response
The manuscript by Parmagnani et al. considers the influence of geomagnetic field and near null magnetic field on photosynthetic apparatus, chloroplast structure, antioxidant proteins, production of peroxides and other. The work performed on high technical level and seems to be very interesting.
R: we thank the reviewer for the appreciation of our work
However, I have some minor remarks:
(1) Please extend technical description of near null magnetic field generation system.
R: we thank the reviewer for this request. Unfortunately, the rules of the journal are strict and do not allow overlapping of language with previous publications. Therefore, we addressed the reader to our previous paper where the full description can be found.
(2) Please add age of plants which were used in experiments.
R: age of plants has been added in the Materials and Methods section
(3) In figure 13, please add direction of changes which were induced by absence of magnetic field.
R: The caption of the figure describes that all the described components of the Z scheme are influenced by the GMF. Adding further graphical direction will probably complicate the scheme. Therefore, we would like to keep it as it is. We thank the reviewer for comprehending the necessity of keeping the scheme as clear as possible.
Reviewer 3 Report
I reviewed the manuscript titled "The Geomagnetic Field (GMF) is required for Lima Bean (Phaseolus lunatus L.) Photosynthesis and Reactive Oxygen Species Production and Modulates MagR Homologs PlIScA2 and PlcpIScA' and found interesting and should be considered for publication after major revision. The following point needs special consideration:
1. Title is very lenghthy with irrelevant information and should be revised.
2. The repitation of double names i.e. Lima bean (Phaseolus lunatus L.) should be avoided. Similar with abbreviations like The geomagnetic field (GMF).
3. Figure 3. Choloroplast morphology needs to be improved.
4. Figure 8 quality needs to be improved. The bands are dispersed and affect the results of the experiment.
Author Response
I reviewed the manuscript titled "The Geomagnetic Field (GMF) is required for Lima Bean (Phaseolus lunatus L.) Photosynthesis and Reactive Oxygen Species Production and Modulates MagR Homologs PlIScA2 and PlcpIScA' and found interesting and should be considered for publication after major revision.
R: we thank the reviewer for the appreciation of our work
The following point needs special consideration:
- Title is very lengthy with irrelevant information and should be revised.
R: the title has been shortened
- The repetition of double names i.e. Lima bean (Phaseolus lunatus L.) should be avoided. Similar with abbreviations like The geomagnetic field (GMF).
R: repetitions have been deleted
- Figure 3. Chloroplast morphology needs to be improved.
R: We uploaded new figures with a higher resolution and dimension
- Figure 8 quality needs to be improved. The bands are dispersed and affect the results of the experiment.
R: Figure 8 entailed the fractionation of native pigment-protein complexes, previously embedded in the thylakoid membranes, upon solubilization with a mild detergent. Non-denaturing gel electrophoresis is known to have a lower resolution than a denaturing SDS-PAGE, because of the charge of surface amino acids, size and configuration of the native complexes, lipid/detergent ratio and other factors. Therefore, this is the best resolution for this technique.